# The Therapeutic Potential of Oral Everolimus for Facial Angiofibromas in Pediatric Tuberous Sclerosis Complex: A Case-Based Analysis of Efficacy

**DOI:** 10.3390/diseases12120334

**Published:** 2024-12-20

**Authors:** George Imataka, Satoshi Mori, Kunio Yui, Ken Igawa, Hideaki Shiraishi, Shigemi Yoshihara

**Affiliations:** 1Department of Pediatrics, Dokkyo Medical University, Tochigi 321-0293, Japan; h-shiraishi@dokkyomed.ac.jp (H.S.); shigemi@dokkyomed.ac.jp (S.Y.); 2Department of Dermatology, Dokkyo Medical University, Tochigi 321-0293, Japan; m-satosh@dokkyomed.ac.jp (S.M.); igawa@dokkyomed.ac.jp (K.I.); 3Department of Pediatrics, Chiba University, Chiba 260-8677, Japan; yui16@bell.ocn.ne.jp

**Keywords:** tuberous sclerosis complex, facial angiofibromas, everolimus, oral stomatitis, sirolimus

## Abstract

Background: Tuberous sclerosis complex (TSC) is an autosomal dominant genetic disorder characterized by mutations in the TSC1 and TSC2 genes, leading to the dysregulation of the mammalian target of rapamycin (mTOR) pathway. This dysregulation results in the development of benign tumors across multiple organ systems and poses significant neurodevelopmental challenges. The clinical manifestations of TSC vary widely and include subependymal giant cell astrocytomas (SEGAs), renal angiomyolipomas (AMLs), facial angiofibromas (FAs), and neuropsychiatric conditions such as autism spectrum disorder (ASD). mTOR inhibitors, notably everolimus, have become central to TSC management, with documented efficacy in reducing the sizes of SEGAs and AMLs and showing promise in addressing additional TSC-related symptoms. Case Presentation: We report the case of an 11-year-old male diagnosed with TSC, presenting with hallmark features including hypopigmented macules, early-onset infantile spasms, SEGA, and AMLs. Initial interventions included adrenocorticotropic hormone (ACTH) therapy and sodium valproate for seizure management and a minimally invasive keyhole craniotomy for SEGA reduction. At age 12, oral everolimus therapy was introduced to address both SEGA recurrence risk and ASD-related social deficits. Over the course of 24 weeks, a reduction in the size and erythema of the patient’s FAs was observed, alongside improvements in social engagement, suggesting potential added benefits of systemic mTOR inhibition beyond tumor control. Results: Treatment with everolimus over a 24-month period led to significant reductions in both FA and AML size, as well as measurable improvements in ASD-associated behaviors. Therapeutic drug monitoring maintained serum levels within the effective range, minimizing adverse effects and underscoring the tolerability and feasibility of long-term everolimus administration. Conclusions: This case underscores the efficacy of oral everolimus in reducing FA size in a pediatric TSC patient, with broader therapeutic benefits that support the potential of mTOR inhibition as a multi-targeted strategy for TSC management. Further studies are needed to explore the full range of applications and long-term impact of mTOR inhibitors in TSC care.

## 1. Introduction

Tuberous sclerosis complex (TSC) represents a multisystem genetic disorder characterized by significant clinical heterogeneity, with an estimated incidence of approximately 1 in 6000 individuals worldwide, affecting diverse populations without a predilection for gender (Northrup et al., 2013) [1]. This relatively high prevalence positions TSC among the more commonly encountered rare genetic disorders. The pathogenesis of TSC is primarily attributable to mutations in either the TSC1 gene on chromosome 9 or the TSC2 gene on chromosome 16, which encode the proteins hamartin and tuberin, respectively. These proteins form a tumor-suppressor complex crucial for the inhibition of the mammalian target of rapamycin (mTOR) pathway, a signaling cascade integral to cellular growth, proliferation, and survival (Crino et al., 2006) [2]. In healthy cells, the hamartin–tuberin complex exerts a negative regulatory effect on the mTOR pathway, ensuring controlled cellular proliferation. However, mutations in TSC1 or TSC2 abrogate this regulatory function, resulting in mTOR pathway hyperactivation, which precipitates the formation of benign tumors, or hamartomas, across various organ systems, including the brain, skin, kidneys, heart, and lungs (Roth et al., 2013) [3].

The clinical manifestations of TSC are protean and widely variable, with benign tumors arising in multiple organs, resulting in a broad spectrum of physical and neurological symptoms. Central nervous system (CNS) involvement, particularly in the form of subependymal giant cell astrocytomas (SEGAs) and cortical tubers, is highly prevalent and correlates with a propensity for epilepsy, one of the most disabling and refractory symptoms associated with TSC. Approximately 80% of individuals with TSC experience seizures, frequently of an early onset and often refractory to conventional antiepileptic therapies, significantly diminishing the quality of life in affected individuals (Franz et al., 2013) [4]. Additionally, TSC frequently manifests with neurodevelopmental and psychiatric comorbidities, including intellectual disabilities, autism spectrum disorder (ASD), and various behavioral disturbances. Studies indicate that ASD is present in approximately 40–50% of individuals with TSC, underscoring the profound neurodevelopmental impact of TSC and the multifaceted challenges it poses (Ehninger and Silva, 2008) [5]. Besides CNS involvement, TSC-related tumors, including renal angiomyolipomas (AMLs) and dermatological lesions such as facial angiofibromas (FAs), further complicate the clinical landscape, contributing to both medical and psychosocial morbidity.

The therapeutic management of TSC has historically been limited to symptomatic treatment, encompassing anticonvulsant drugs for seizure control, surgical intervention for tumors with immediate clinical risk, and supportive strategies for neuropsychiatric symptoms. However, advancements in the understanding of TSC genetics and cellular pathophysiology have paved the way for molecularly targeted therapies, particularly those that address the aberrant mTOR pathway activity central to TSC pathogenesis. The development of mTOR inhibitors has introduced a novel therapeutic avenue that directly targets the hyperactive mTOR signaling resulting from TSC1 and TSC2 mutations. Everolimus, an orally bioavailable mTOR inhibitor, was approved in 2013 for use in the treatment of TSC-related SEGAs and AMLs based on robust evidence from clinical trials demonstrating its efficacy in reducing tumor volume and delaying surgical interventions (Davies et al., 2017) [6].

Pivotal studies, including the EXIST-1 and EXIST-2 trials, provided compelling evidence supporting the efficacy of everolimus in TSC management by demonstrating significant tumor regression in both SEGA and AML cases (Bissler et al., 2016; Kingswood et al., 2014) [7,8]. The therapeutic impact of everolimus has been particularly transformative for individuals with SEGAs, where tumor size reduction mitigates the risk of life-threatening complications, such as obstructive hydrocephalus, without necessitating invasive surgical procedures. Beyond tumor reduction, emerging data suggest that mTOR inhibitors may also ameliorate certain neuropsychiatric manifestations of TSC. Preliminary studies have observed that everolimus may modulate social and cognitive impairments, particularly in patients presenting with ASD-related symptoms (Wataya-Kaneda et al., 2017; Krueger et al., 2013) [9,10]. Although interindividual variability in response is notable, these findings are significant as they highlight the potential role of mTOR dysregulation in the neurodevelopmental and behavioral phenotypes associated with TSC, warranting further investigation into mTOR inhibition as a strategy for addressing neuropsychiatric symptoms.

Despite these advancements, the therapeutic landscape of TSC remains complex. Not all individuals with TSC respond uniformly to mTOR inhibitors, and adverse effects, including immunosuppression and stomatitis, may impact treatment adherence. Furthermore, the long-term implications of sustained mTOR inhibition are not yet fully elucidated, underscoring the necessity for continued clinical surveillance and further research. Future investigations are expected to elucidate the parameters within which mTOR inhibition may optimally address neuropsychiatric symptoms and to explore combinatory approaches that may enhance treatment efficacy and mitigate adverse outcomes.

In this report, we examine the effects of everolimus treatment in an adolescent male with TSC, with a particular focus on the therapeutic impact on SEGA, AML, and FAs, as well as the observed improvements in ASD-related social behaviors. Given the prevalence of FAs and the psychosocial challenges they impose, this case underscores the broader therapeutic potential of systemic mTOR inhibition, suggesting that everolimus may contribute to improved physical and neurodevelopmental outcomes in TSC.

## 2. Case Report

The patient, an 11-year-old male, presented with clinical findings consistent with a diagnosis of TSC, identified initially in infancy following the observation of multiple hypopigmented macules, or “ash leaf” spots, on the skin. These lesions led to further genetic evaluation and imaging studies, confirming the TSC diagnosis through cutaneous and neurological markers. At 13 months, the patient developed infantile spasms, a form of epilepsy often associated with TSC, managed with adrenocorticotropic hormone (ACTH) therapy at 0.015 mg/kg/day for 2 weeks and sodium valproate 20 mg/day, leading to effective seizure control. By the age of 8, a brain MRI revealed a SEGA in the right lateral ventricle and bilateral renal AMLs, consistent with the multi-organ involvement typical of TSC.

At the age of 10, comprehensive cognitive assessments revealed an intelligence quotient (IQ) of 67, indicative of significant deficits in adaptive functioning and social cognition, consistent with the neurodevelopmental profile often observed in TSC.

At the age of 11, the patient underwent a minimally invasive keyhole craniotomy to achieve a partial resection of the SEGA, a procedure performed to mitigate the risk of obstructive hydrocephalus associated with tumor growth. Persistent impairments in social interaction and communication subsequently led to a formal diagnosis of autism spectrum disorder (ASD), a neurodevelopmental condition frequently linked to TSC due to the dysregulation of the mTOR signaling pathway.

Following formal approval by the Medical Ethics Committee of Dokkyo Medical University (approval No. 27014) and upon obtaining informed consent from the patient’s mother, oral administration of everolimus was initiated when the patient reached the age of 12. This therapeutic intervention aimed to address the dual concerns of preventing the recurrence of SEGAs and potentially mitigating autism spectrum disorder (ASD)-related symptoms, given the established role of mTOR inhibition in modulating both tumorigenic and neurodevelopmental processes. The decision to commence everolimus treatment was based on its demonstrated efficacy in reducing SEGA progression risk in TSC patients, while also taking into consideration emerging evidence of its beneficial effects on neuropsychiatric symptoms associated with mTOR dysregulation.

The initial prescribed dose of everolimus was 4.4 mg per day. Thereafter, the patient was treated a maintenance dose of 6 mg/day. Prior to the commencement of treatment, multiple prominent facial angiofibromas (FAs) were observed around the patient’s nose and cheeks, presenting as characteristic red papules associated with tuberous sclerosis complex and contributing to notable cosmetic and psychosocial concerns (Figure 1A,B).

Several days following the initiation of everolimus therapy, the patient developed multiple mouth ulcers, which were identified as likely manifestations of stomatitis, a known adverse effect associated with mTOR inhibitors. These lesions were effectively managed through diligent oral hygiene practices, leading to their resolution without further complications or the necessity for additional pharmacologic intervention (Figure 2). Periodic dose adjustments were implemented based on therapeutic drug monitoring to ensure optimal efficacy, with serum everolimus levels carefully maintained within a therapeutic range of 7.32 to 18.1 ng/mL. By three months after the initiation of treatment, a discernible trend of improvement in the patient’s facial angiofibromas (FAs) began to emerge, indicating a positive response to systemic mTOR inhibition (Figure 1C,D). By six months into the treatment regimen, both the size and erythema of the facial angiofibromas had visibly diminished, indicating a noteworthy and somewhat unexpected therapeutic response of these lesions to systemic mTOR inhibition with everolimus (Figure 1E). The patient continued treatment with oral everolimus and, over the months, showed improvement in social behavior, including improved eye contact. A follow-up MRI scan showed a reduction in renal AMLs after 24 months. The patient also continued to show improvement in facial skin FAs, demonstrating the continued efficacy of everolimus in the management of TSC-related symptoms. The patient continues to receive oral everolimus therapy. In this case, his facial angiofibromas (FAs) improved with oral everolimus alone, and we did not use rapamycin gel.

A/B: Pre-treatment facial photographs

In the pre-treatment facial photograph, prominent red skin nodules and areas of diffuse blue tint are symmetrically distributed over the entire nose and extend into other facial areas. These nodules are very prominent and stand in marked contrast to the surrounding skin.

C/D: 3 months after treatment

Three months after the start of treatment, the nodules in the nose, along with similar nodules in other facial areas, show a subtle but observable reduction in size. They are flatter and less noticeable. Additionally, the previously prominent red skin lesions on the face have almost completely disappeared. This favorable change motivated the patient to manage own medication regimen, and adherence to his oral medications improved markedly.

E: 6 months after everolimus treatment

Six months after the start of everolimus treatment, the remaining nodules on the facial skin have further reduced in size and continue to flatten.

The patient’s oral mucosa showed characteristic signs of stomatitis, including marked mucosal pain, which appeared shortly after the start of everolimus treatment. This initial symptom is consistent with common side effects associated with mTOR inhibitors, particularly everolimus, which often induce oral discomfort and ulcers due to their effect on mucosal integrity. The patient’s symptoms improved only with oral care, such as brushing the teeth, disinfecting the mouth, and gargling.

## 3. Discussion

TSC-related gene mutations lead to PI3K/AKT/mTOR pathway dysregulation, promoting benign tumor growth and neurodevelopmental issues (Huang et al., 2008) [11]. Everolimus, a mechanistic target of the rapamycin complex 1 (mTORC1) inhibitor, effectively reduces SEGA and AML size, as established in the EXIST trials (Franz et al., 2013) [4]. Additionally, evidence indicates that everolimus may improve ASD symptoms in TSC, possibly by modifying mTOR-driven neurodevelopmental processes (Wang et al., 2015) [12]. Animal models suggest that mTORC1 inhibition could reverse ASD-like behaviors, with clinical reports supporting these findings in human TSC cases (Yui et al., 2019) [13].

### 3.1. Everolimus for Facial Angiofibromas

Facial angiofibromas (FAs) are highly prevalent in TSC, often exacerbating social stigma (Koenig et al., 2018) [14]. Systemic everolimus treatment provided notable FA improvement in this case, suggesting that mTOR inhibition can target both dermatologic and neurodevelopmental symptoms. While rapamycin gel has shown efficacy for FA as a topical mTOR inhibitor, systemic therapy may offer more comprehensive benefits across TSC manifestations (Pass et al., 2021) [15]. Careful monitoring for side effects such as stomatitis is essential in systemic treatment (Devaraj et al., 2019) [16].

### 3.2. Topical vs. Systemic mTOR Inhibition

Topical rapamycin (sirolimus) gel offers localized FA management with minimal systemic absorption, though it requires caution with sun exposure due to photosensitivity (Hart et al., 2021) [17]. By contrast, systemic everolimus addresses multiple TSC symptoms but necessitates careful monitoring to manage adverse effects. Combining both approaches could optimize outcomes in TSC, balancing localized and systemic benefits (Koenig et al., 2018) [18]. Safety and efficacy studies on mTOR inhibitors, particularly in pediatric populations, emphasize the need for personalized approaches to treatment (Segal et al., 2020) [19]. Table 1 summarizes the characteristics of oral everolimus and ointment sirolimus. Several studies have reported photosensitivity in patients using everolimus, suggesting an increased risk of skin reactions, particularly due to immunosuppression. However, the risk of photosensitivity is not significantly higher than that with sirolimus, and its systemic effects as an oral medication differ from those of topically used sirolimus (Liu M et al., 2024; Lin Y-T et al., 2022) [20,21].

### 3.3. Future Directions in TSC Treatment

The limitations of everolimus, particularly the common adverse effects such as stomatitis and hyperlipidemia, underscore the pressing need for ongoing research into more refined and targeted therapeutic options for TSC management. These adverse events can affect patient compliance and limit the utility of current mTOR inhibitors, especially in long-term treatment regimens, thereby emphasizing the importance of exploring alternative or adjunctive therapeutic approaches. Advances in pharmacological research have introduced dual mTORC1 mechanistic target of rapamycin complex 2 (mTORC2) inhibitors, which hold promise for the more comprehensive modulation of mTOR signaling by simultaneously targeting multiple aspects of the pathway. This dual inhibition approach may provide broader and more effective control over the spectrum of TSC manifestations, addressing both cellular proliferation and neurodevelopmental symptoms with greater precision (Martinez-Aguayo et al., 2020) [22]. Moreover, the development of novel mTOR-targeting therapies, including combination regimens that pair mTOR inhibitors with other molecular agents, represents a promising direction to enhance therapeutic efficacy while potentially mitigating adverse effects. Such approaches are currently under investigation and could significantly advance treatment paradigms in TSC, ultimately offering more personalized and effective interventions for patients (Medley MA et al., 2022; Lendvai et al., 2019; Hussain et al., 2020; Bar-Peled et al., 2019; Davies DM et al., 2019) [23,24,25,26,27].

## 4. Conclusions

This case highlights the multifaceted efficacy of everolimus in the management of tuberous sclerosis complex (TSC), demonstrating its therapeutic impact on both physical manifestations, such as tumor size reduction, and neurodevelopmental symptoms, including improvements in social behavior. The findings suggest that everolimus, as an mTOR inhibitor, plays a crucial role in addressing the diverse and complex symptomatology of TSC. Looking forward, continued research into the safety, long-term effects, and broader applications of mTOR inhibitors will be essential for the optimization of treatment strategies.

## Figures and Tables

**Figure 1 diseases-12-00334-f001:**
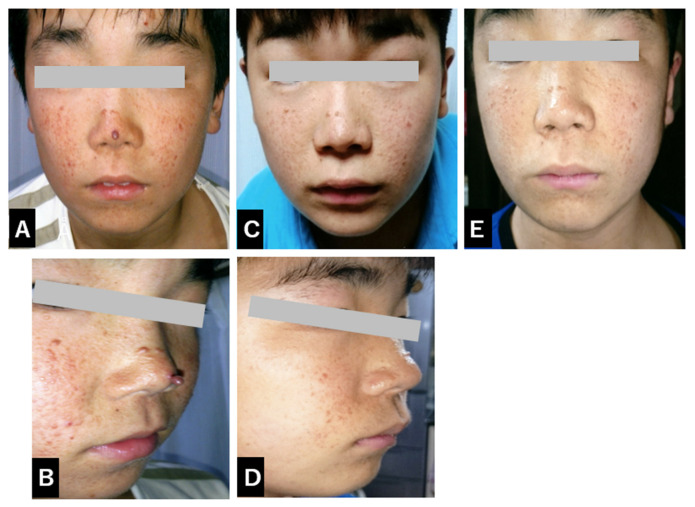
Sequential changes in facial appearance and angiofibroma development (**A**–**E**).

**Figure 2 diseases-12-00334-f002:**
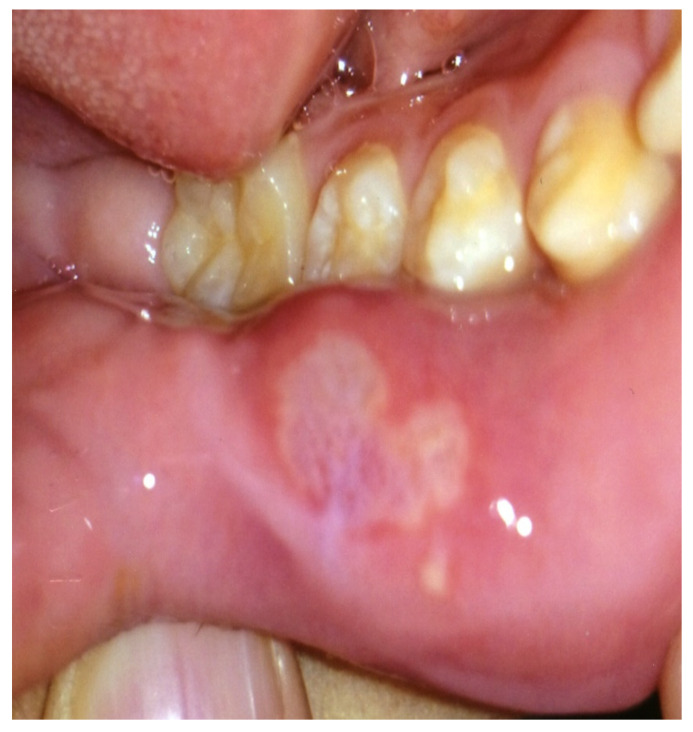
Detailed examination of the condition of the patient’s oral mucosa during treatment.

**Table 1 diseases-12-00334-t001:** Detailed pharmacological and medical comparison of everolimus and sirolimus.

Category	Everolimus (Oral)	Sirolimus (Topical)
Formulation	Oral tablet (oral medication)	Topical medication (topical ointment)
Mechanism of Action	Everolimus selectively inhibits mTORC1 (mechanistic target of rapamycin complex 1), suppressing cell cycle progression. mTORC1 is involved in regulating cell growth, proliferation, survival, and metabolism, and its inhibition suppresses the growth of tumor cells and abnormal cells. It is particularly effective for SEGA (subependymal giant cell astrocytoma), skin angiofibromas, and kidney tumors related to TSC.	Sirolimus locally inhibits mTORC1, suppressing the excessive growth of skin cells and angiogenesis, contributing to the improvement of facial angiofibromas and skin rashes. Being localized, its systemic effects are minimal. The inhibition of mTORC1 primarily suppresses cell proliferation and angiogenesis, particularly the thickening of the skin epidermis.
Indications	Everolimus is used in the treatment of kidney, brain (SEGA), and skin lesions associated with tuberous sclerosis complex (TSC). It is particularly effective for renal tumors and facial angiofibromas, and used for lung lesions in TSC.	Sirolimus is used in the treatment of facial angiofibromas and skin lesions associated with TSC. Aimed at improving facial skin lesions locally.
Pharmacological Effects	Everolimus controls cell growth, proliferation, and metabolism by inhibiting mTORC1, particularly suppressing tumor and abnormal cell proliferation. This leads to tumor size reduction and growth suppression. Efficacy has been demonstrated in shrinking renal tumors, brain tumors (SEGAs), skin angiofibromas, and lung nodules.	Sirolimus locally inhibits mTORC1, leading to an improvement in skin angiofibromas and rashes. Its localized action minimizes the risk of systemic side effects. The main effect is the suppression of cell division and angiogenesis at the skin surface.
Administration	Everolimus is taken orally, with a recommended daily dose, unaffected by meals. Peak plasma concentration occurs 1–2 h after administration, with a half-life of approximately 30 h. It is metabolized primarily in the liver, with metabolites excreted in the urine.	Sirolimus is applied topically, usually twice daily. The medication works locally on the skin, with minimal systemic effects. Blood concentrations are kept low.
Pharmacokinetics	Everolimus is absorbed from the gastrointestinal tract and metabolized in the liver by the CYP3A4 enzyme system. Its plasma concentration may fluctuate due to food intake, so regular monitoring is recommended. It is excreted after liver metabolism, with a half-life of about 30 h, requiring consistent dosing intervals.	After topical application, sirolimus acts locally with extremely low systemic absorption, minimizing the risk of systemic side effects. Data on local pharmacokinetics are limited, but the drug acts on the skin’s surface, maintaining localized effects.
Side Effects	Immunosuppressive effects increase the risk of infections (particularly pneumonia and urinary tract infections). Gastrointestinal symptoms (mouth ulcers, diarrhea), liver dysfunction, increased blood sugar, and hypertension are reported. Skin rashes, edema, and respiratory symptoms (coughing, shortness of breath) may also occur.	Local skin side effects (dryness, itching, redness, inflammation) are primarily observed. Systemic side effects are rare, but occasionally, skin hypersensitivity or allergic reactions are reported.
Clinical Trial Results	Everolimus has shown clear efficacy in treating kidney tumors, brain tumors (SEGAs), and skin lesions in TSC, especially in tumor shrinkage and disease progression inhibition. Long-term clinical trials have confirmed its therapeutic effectiveness.	Sirolimus has demonstrated localized effectiveness for facial angiofibromas, showing improvement in facial rashes and blood vessel tumors. Local treatment leads to visible improvements, with mild side effects, making it a favorable option for patients.

## Data Availability

Data are contained within the article.

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
