# Peer review of "The Therapeutic Potential of Oral Everolimus for Facial Angiofibromas in Pediatric Tuberous Sclerosis Complex: A Case-Based Analysis of Efficacy"

_diseases, 2024, doi:10.3390/diseases12120334_

Round 1
Reviewer 1 Report
Comments and Suggestions for Authors
This is a well-written case report of a boy with Tuberous Sclerosis Complex Disease with up-to-date comments on therapy.
minor remark:
line 177: "her" probably should read "his"
Author Response
Comments and Suggestions for Authors
This is a well-written case report of a boy with Tuberous Sclerosis Complex Disease with up-to-date comments on therapy.
minor remark:
line 177: "her" probably should read "his"
Our response
Thank you for your detailed review of my paper. I have corrected my error “her” to “his”.
Reviewer 2 Report
Comments and Suggestions for Authors
The author presented a case of a boy with tuberous sclerosis complex whose facial angio-fibroma, among others, improved with everolimus medication.
The case was clearly presented. The presentation provides further evidence of the efficacy of systemic mTOR inhibitor, everolimus.
However, the manuscript would be more interesting to the readers if the authors could find a new perspective in the case presentation.
Author Response
Comments and Suggestions for Authors
The author presented a case of a boy with tuberous sclerosis complex whose facial angio-fibroma, among others, improved with everolimus medication.
The case was clearly presented. The presentation provides further evidence of the efficacy of systemic mTOR inhibitor, everolimus.
However, the manuscript would be more interesting to the readers if the authors could find a new perspective in the case presentation.
Our response
I greatly appreciate your important comments. I believe this manuscript will be more interesting to readers if they can find new perspectives from this case report. Therefore, we have created and added Table 1, which clarifies the differences between the two drugs. The addition of this table makes this a detailed and scholarly case report. Thank you very much.
Reviewer 3 Report
Comments and Suggestions for Authors
The publication addresses the influence of everolimus in particular on angiofibromas of the skin in patients with tuberous cerebral sclerosis. In a small group of childhood patients, there was a significant improvement in the findings. The side effects were manageable. At the same time, SEPA also improved. In addition, the influence on neurological complications of the disease was favorable. Since angiofibromas are stigmatizing for those affected, successful therapy is crucial for social integration. So far, little has been published about this aspect of the disease. A case report with a local therapy should be published.
Author Response
Comments and Suggestions for Authors
The publication addresses the influence of everolimus in particular on angiofibromas of the skin in patients with tuberous cerebral sclerosis. In a small group of childhood patients, there was a significant improvement in the findings. The side effects were manageable. At the same time, SEPA also improved. In addition, the influence on neurological complications of the disease was favorable. Since angiofibromas are stigmatizing for those affected, successful therapy is crucial for social integration. So far, little has been published about this aspect of the disease. A case report with a local therapy should be published.
Our response
We greatly appreciate your comments on our paper. Based on the comments of the other three reviewers, we have revised our paper to make it better. We are very appreciative of the reviewers' efforts.
Reviewer 4 Report
Comments and Suggestions for Authors
The paper describes a promising treatment for tuberous sclerosis in reducing not only angiomas but possibly also other symptoms associated with the disease.
The paper is well-written and of high interest.
I have only very minor comments:
1. A topical cream and ointment formulations of mTOR inhibitors for facial angiomas are also available. Would you mind mentioning this and whether your patient ever used a topical formulation at any point?
2. Please include dosing regimens when describing the treatments given to the patient (ACTH, na valproat, initial calculations of chosen dose for everolimus).
3. Please make sure that all abbreviations are given in full when first mentioned (both in the abstract and text (ACTH, mTORC1, mTORC2)
4. Oral everolimus is also known to cause photosensitivity. Although less severe, the photosensitivity is generalized to all exposed skin. When you mention that topical formulations have the disadvantage of photosensitivity, perhaps mention that systemic intake does not eliminate this side effect completely.
I do not see a blue highlighted area of the nose. The photos have no highlighted area.
Page 4, lines 177 and 178 - you write 'her', but the patient is a male.
Author Response
Comments and Suggestions for Authors
The paper describes a promising treatment for tuberous sclerosis in reducing not only angiomas but possibly also other symptoms associated with the disease.
The paper is well-written and of high interest.
I have only very minor comments:
- A topical cream and ointment formulations of mTOR inhibitors for facial angiomas are also available. Would you mind mentioning this and whether your patient ever used a topical formulation at any point?
Our response
In our case, we used only everolimus, an mTOR inhibitor for TSC. No other creams or ointments were used.
So I added the sentence “In this case, his facial angiofibromas (FAs) improved with oral everolimus alone, and we did not use rapamycin gel.” in the text.
- Please include dosing regimens when describing the treatments given to the patient (ACTH, na valproat,
initial calculations of chosen dose for everolimus).
Our response
Thank you for your remarks. I have added the initial dosages of, ACTH at 0.015 mg/kg/day for 2 weeks and and sodium valproate 20mg/day and select doses of everolimus 6mg/day to the text to be more accurate. In addition, add the maintenance dose as follows; “Thereafter, the patient was followed at a maintenance dose of 6 mg/day.”
- Please make sure that all abbreviations are given in full when first mentioned (both in the abstract and text (ACTH, mTORC1, mTORC2)
Our response
I have added the formal names of the three medical terms “ACTH”, “mTORC1”, and “mTORC2” to the text.
- Oral everolimus is also known to cause photosensitivity. Although less severe, the photosensitivity is generalized to all exposed skin. When you mention that topical formulations have the disadvantage of photosensitivity, perhaps mention that systemic intake does not eliminate this side effect completely.
Our response
I appreciate your important point that oral everolimus is also known to cause photosensitivity. I have added two citations “20” and “21” regarding this point. I have also added a discussion to the text. Thank you for your important information.
I do not see a blue highlighted area of the nose. The photos have no highlighted area.
Our response
Thank you for your detailed review of my paper. The “blue highlighted area of the nose” in my text was a typo. It has been deleted.
Page 4, lines 177 and 178 - you write 'her', but the patient is a male.
Our response
Thank you for your detailed review of my paper. I have corrected my error “her” to “his”.